# The Emerging Role of Chromatin Remodeling Complexes in Ovarian Cancer

**DOI:** 10.3390/ijms232213670

**Published:** 2022-11-08

**Authors:** Ieva Vaicekauskaitė, Rasa Sabaliauskaitė, Juozas Rimantas Lazutka, Sonata Jarmalaitė

**Affiliations:** 1Laboratory of Genetic Diagnostic, National Cancer Institute, Santariškių 1, LT-08406 Vilnius, Lithuania; 2Institute of Biosciences, Vilnius University, Sauletekio Avenue 7, LT-10222 Vilnius, Lithuania; 3Laboratory of Clinical Oncology, National Cancer Institute, Santariškių 1, LT-08406 Vilnius, Lithuania

**Keywords:** chromatin remodeling complexes, ovarian cancer, ARID1A, SWI/SNF, ISWI, CHD, INO80

## Abstract

Ovarian cancer (OC) is the fifth leading cause of women’s death from cancers. The high mortality rate is attributed to the late presence of the disease and the lack of modern diagnostic tools, including molecular biomarkers. Moreover, OC is a highly heterogeneous disease, which contributes to early treatment failure. Thus, exploring OC molecular mechanisms could significantly enhance our understanding of the disease and provide new treatment options. Chromatin remodeling complexes (CRCs) are ATP-dependent molecular machines responsible for chromatin reorganization and involved in many DNA-related processes, including transcriptional regulation, replication, and reparation. Dysregulation of chromatin remodeling machinery may be related to cancer development and chemoresistance in OC. Some forms of OC and other gynecologic diseases have been associated with mutations in specific CRC genes. Most notably, *ARID1A* in endometriosis-related OC, *SMARCA4*, and *SMARCB1* in hypercalcemic type small cell ovarian carcinoma (SCCOHT), *ACTL6A*, *CHRAC1*, *RSF1* amplification in high-grade serous OC. Here we review the available literature on CRCs’ involvement in OC to improve our understanding of its development and investigate CRCs as possible biomarkers and treatment targets for OC.

## 1. Introduction

### 1.1. Ovarian Cancer Types and Mutations

Ovarian cancer (OC) is the third most common gynecologic malignancy and the second deadliest oncogynecologic disease in the world [1]. It is estimated that just in the United States, more than 12.8 thousand women will die from OC this year [2]. Early-stage OC presents a good diagnosis and a more than 90% 5-year survival rate. However, about 80% of OC patients are diagnosed with stage III-IV disease, where survival, even after therapy, is only 30% [3]. The high OC frequency and mortality rates are attributed to the lack of specific symptoms and insufficiency of the main OC blood biomarker CA125 to reduce mortality. Due to insufficient prognostic power, the CA125 serum biomarker is recommended as a prognostic tool only in high-grade ovarian cancer (HGSOC) patients [4]. Currently, there are no sensitive molecular biomarkers of clinical value for OC screening and patient management. Generally, after diagnosis, OC patients are treated by bilateral salpingo-oophorectomy (surgical removal of ovaries and fallopian tubes) with platinum or taxane therapy. Other therapies are available only after chemoresistance development which almost inevitably occurs in most HGSOC cases [5].

One of the culprits of OC chemoresistance and early therapy failure is the heterogeneity of the disease. Histologically, OCs are subdivided into epithelial and non-epithelial tumors. The non-epithelial OC only comprises 10% of OC cases. These are malignant germ cell tumors (e.g., small cell carcinoma of the ovary) that are usually present in preadolescent women, and sex cord-stromal tumors (granulosa cell tumors, fibroma, Sertoli-Leydig cell tumors, and others) that are more common and typically affect postmenopausal women [6]. Due to its rarity, non-epithelial tumors are rarely studied, and most research focuses on epithelial OC.

Epithelial type of OC is further subdivided into five major types according to histologic differences: the endometrioid, clear cell, mucinous, low-grade, and high-grade serous OCs [7]. According to the dualistic ovarian cancer model, epithelial OCs are grouped into two major types based on the site of origin of the tumor [8]. Type II tumors comprise mainly HGSOC but can also include carcinosarcomas and undifferentiated carcinomas. Unique for its high frequency of *TP53* mutations and chromosomal instability, these tumors arise from the intraepithelial carcinoma in the fallopian tube. However, other origins, such as solid pseudoendometrioid transition tumors or borderline or low-grade serous OC, are possible. All other epithelial tumors (low-grade serous, mucinous, clear cell, borderline, seromucinous, and endometrioid) are considered less of a concern due to their slow-growing nature. Although borderline, mucinous, and seromucinous tumor origin is unclear, the endometroid and clear cell tumors are thought to originate from endometriosis and are similar to endometrial cancers [9]. Albeit highly heterogeneous, mutations in *PIK3CA*, *PTEN*, *KRAS*, *BRAF*, *CTNNB1*, and *ARID1A* genes are frequent amongst type I epithelial OCs and are considered potential OC biomarkers [10].

Regardless of type, a third of OC cases are related to homologous recombination gene insufficiency [5]. About 23% of OC cases are inherited. 10–15% of inherited cases are associated with *BRCA1* and *BRCA2* germline mutations [11,12]. The rest of the inherited OC cases are caused by *TP53* mutations due to patients with the Li-Fraumeni syndrome [13]. The second gene group, most often mutated in OC, are the genes encoding proteins that are involved in chromatin remodeling complexes (CRCs) which are crucial for transcription, DNA repair, and most other cell day-to-day fluctuations in chromatin availability states.

### 1.2. Chromatin Remodelers and Their Functions in Human Cancer

Inside every human cell’s nucleus fits the genome comprising three billion DNA base pairs, almost two meters in length. This is achieved by winding the DNA around histone protein complexes forming nucleosomes that are the building blocks of the chromatin nucleoprotein structure. Each nucleosome has an octamer core made of H2A, H2B, H3, and H4 histone protein pairs and fits 147 bp of DNA. The nucleosomes are connected with linker DNA of 10–60 bp in length [14] and form the chromatin complex, a functionally dynamic structure. During interphase, chromatin hierarchically organizes and compartmentalizes DNA into active (euchromatin) and silent (heterochromatin) regions in the nucleus [15]. The distinct chromatin states are achieved through epigenetic modifications of the DNA, histone N-terminal tails, and nucleosome management done by ATP-dependent chromatin remodeling protein complexes. CRCs aid in nucleosome assembly and maintain dynamic chromatin activity through sliding, rearranging, evicting, and modifying nucleosomes. CRCs achieve these complex functions using their ATPase domain to translocate DNA in regard to histones [16]. Chromatin remodelers are involved in most essential cell processes, most notably regulating RNA pol II activity, gene silencing, homologous recombination (HR), and DNA repair [17].

There are four conserved families of ATP-dependent chromatin remodelers in mammals: chromodomain helicase DNA binding (CHD), imitation switch (ISWI), inositol requiring 80 (INO80), and switch/sucrose non-fermenting (SWI/SNF) [16]. The multiprotein complexes that remodel chromatin all share a common core ATPase-translocase subunit from the RNA/DNA helicase superfamily 2 (Snf2), but differ in accessory proteins that read epigenetic modifications and regulate the enzymatic activity of the complexes. SWI/SNF complexes have bromodomains capable of reading acetylated lysine on histone tails and function as chromatin accessibility regulators by sliding and evicting nucleosomes. ISWI reader domain (C-terminal HAND-SANT-SLIDE) binds to unmodified H4 tails and promotes chromatin assembly as well as proper nucleosome spacing. CHD remodelers are known to have similar functions to ISWI, although they contain two tandem chromodomains that read H3K4me3 active chromatin marks and are also known to bind to AT-rich DNA motifs. INO80 family remodelers are known to act in recombination and DNA replication as it is capable of binding to specialized DNA structures. In addition to nucleosome spacing, the INO80 family facilitates histone exchange by replacing nucleosome dimers and regulating histone variants [18].

Traditionally, SWI/SNF chromatin remodelers are associated with increasing chromatin accessibility, INO80 mainly acts as a chromatin repressor, and ISWI and CHD are most often described as histone chaperones in chromatin assembly [19]. However, these broad classifications are not fixed, as particular chromatin remodelers can be classified into two groups based on the chromatin regions they occupy. BRG1 (SWI/SNF), SNF2H (ISWI), CHD3, and CHD4 proteins are mainly found in active chromatin regions, while BRM (SWI/SNF), INO80, SNF2L (ISWI), CHD1 are associated with inactive chromatin [20].

#### 1.2.1. SWI/SNF Complex Members and Functions

SWI/SNF was the first CRC discovered and, thus, is the most well-studied. The complex gets its SWI/SNF (mating Type SWItch/Sucrose Non-Fermentable) name from the yeast where it was first discovered [21]. The Brg/Brahma-associated factors (BAF), as they are known in mammals, are ATP-dependent CRCs responsible for nucleosome repositioning by nucleosome sliding and eviction, resulting primarily in increased chromatin availability. Besides nucleosome relocation at promoter and enhancer regions, SWI/SNF also facilitates transcription factor binding, histone-modifying enzyme recruitment, and regulation of chromatin looping to facilitate enhancer and promoter interaction [22].

The final assembled SWI/SNF complexes are ∼1 to 1.5 MDa size, 10–15 protein conglomerates subcategorized into three groups: canonical BRGI/BRM-associated factor (cBAF), polybromo-associated BAF (PBAF), and recently characterized non-canonical, GLTSCR1/L-associated BAFs (ncBAF or GBAF) [23]. Each complex contains a common core and ATPase cap that always includes ether BRG1 (encoded by *SMARCA4*) or BRM (*SMARCA2*) catalytic subunits. The different SWI/SNF complexes are distinguished by complex-specific accessory subunits (Table 1). cBAF non-core subunits include ARID1A/ARID1B and DPF1/2/3, PBAF has PBRM1, PHF10, ARID2, and BRD7, and ncBAF contains GLTSCR1/GLTSCR1L and BRD9 complexes. Although some of the SWI/SNF domains have known tasks, e.g., bromodomain-containing PBRM1 and BRD7 subunits that read histone acetylation marks such as active chromatin mark H3K27Ac, functions of other non-catalytic subunits are less understood [24]. Moreover, cBAF, PBAF, and ncBAF complexes are preferably assembled on different chromatin sites. The ncBAFs are uniquely localized on CTCF sites, especially ones co-localized with H3K4me1 primed sites, while PBAFs occupy gene body and active promoters (H3K27ac, H3K4me3), and cBAFs are more often localized on distal sites, such as active enhancers (H3K27ac, H3K4me1) [25].

SWI/SNF complexes are well known for their essential functions in many pathways related to cancerogenesis. Most notably, loss of SWI/SNF or its subunits is known to impair DNA damage repair, cell cycle regulation, critical signaling pathways, such as PI3K/AKT/mTOR, and MYC regulation (described in detail below).

#### 1.2.2. ISWI Complex Members and Functions

ISWI (Imitation switch) is a highly conserved family of chromatin remodelers. ISWI contains SWI2/SNF2 family ATPase domain, a part of a broader superfamily of DEAD/H-helicases, a specific HAND-SANT-SLIDE DNA, and histone-tail binding domain. The human ISWI has either SNF2L (*SMARCA1*) or SNF2H (*SMARCA5*) ATPases that, together with regulatory domains, form up to 16 CRCs with various functions: ACF, CHARC, RSF perform nucleosome spacing, WICH is primarily involved in replication by incorporating histones into newly synthesized DNA, NoRC perform transcriptional repression, NURF and CERF are responsible for transcriptional activation [26,27]. Many ISWI family CRCs are involved in DSB either by translocating nucleosomes or by recruitment of DSB machinery through protein interactions. Acf1 has been found to interact directly with Ku70/80 to promote non-homologous end joining (NHEJ) repair. Meanwhile, ACF, RSF, NURF, CERF, NoRC, and WICH are also known to regulate transcription under different circumstances due to their influence on RNA polymerases [28].

#### 1.2.3. CHD Family Remodelers

The chromodomain-helicase-DNA binding (CHD) family consists of nine (CHD1-9) proteins. The CHD family shares a highly conservative helicase-ATPase domain from the SWItch2/SNF2 superfamily of ATPases and two tandem chromatin organization modifier domains (chromodomains), that read histone modifications. In particular, CHD1 binds to the H3H4me mark and CHD5 to H3K27me3 [29]. CHD proteins are subdivided into three subfamilies based on the other accessory domains. Subfamily I (CHD1, CHD2) bind to AT-rich DNA motifs through its C-terminus, subfamily II (CHD3-5) has PHD zinc finger domain in its N-terminal region and binds to methylated or acetylated histone residues, subfamily III (CHD6-9) is the most variable family, binding to non-modified DNA through SAINT and BRK domains. Although CHD’s primary function is nucleosome (dis)assembly, CHD proteins are involved in different stages of transcription: CHD7 and CHD8 are associated with transcription activation while CHD1 are more associated with termination [30].

CHD3 and CHD4 proteins are also included in the nucleosome remodeling and histone deacetylase complex (Mi-2/NuRD) that also contains histone deacetylases (HDAC1 and HDAC2), and other proteins (MBD2, MBD4, MTA1-3, GATAD2A/B, RBBP4/7) that bind to methylated DNA or histone tails [31]. Generally, CHD3 or CHD4 containing NuRD complexes are associated with transcriptional repression. However, new studies have found that NuRD also governs transcriptional activation, replication, and DNA repair, adding to NuRD’s involvement in cancer progression [32]. Although CHD5 is not a part of the NuRD complex, immunoprecipitation analysis found CHD5 to bind with most of NuRD’s components, including MTA1/2, GATAD2A, HDAC1/2, RBBP4/7, and MBD2/3, forming a similar complex to NuRD [33].

The first CHD gene linked with human cancer was *CHD5*, with the discovery that its locus 1p36.31 is often deleted in glioma and other tumors [34]. Curiously, CHD5 expression is low in most tissues except the nervous system and testis. However, in other tissues, CHD5 expression is induced by DNA damage [29]. CHD5 expression is decreased by lysine demethylase JMJD2A/KDM4A leading to increased cell senescence due to p53 pathway dysregulation [35]. CHD5 regulates INK4A/ARF locus that codes p19(Arf) and other genes that stabilize p53; thus, CHD5 loss causes p53 deficiency [34]. Moreover, low CHD5 expression correlates with the upregulation of DNA damage response (DDR) markers γ-H2AX and CHK2 Thr68 phosphorylation in nuclear loci of pancreatic cancer cells [36].

#### 1.2.4. INO80 Family

Inositol requiring 80 (INO80) family, named after inositol-responsive gene expression regulation [37], is the newest member of CRCs. Humans consist of INO80, SRCAP, and p400/TIP60 complexes. Like SWI/SNF, INO80 family complexes are large, 14–15 protein conglomerations. The defining feature of the INO80 family is the insertion of the spacer region in its SNF2 helicase that interacts with RuvB-like helicases and actin-related proteins (Arps). As RuvB is a bacterial DNA repair factor, RuvB-like subunits suggest INO80’s role in DNA repair [38]. Although the primary biochemical function of INO80 is nucleosome sliding, during DNA repair, TIP60 acetylates H4 and H2A.Z histones, which stimulates incorporation of H2A.Z:H2B dimers by SRCAP, while INO80 reverses this action [39]. INO80 is recruited to the damage site via association with γ-H2AX (phosphorylated H2AX) histone mark to help with nucleosome eviction during HR [38].

Just like other CRCs, INO80 also governs transcription. INO80 can activate mRNA transcription by −1 and +1 nucleosome sliding at transcription start sites where it is recruited by RNAPII and transcription elongation complex PAF1, however, at heterochromatin sites INO80 silences transcription by removing the H2A.Z mark, preventing H3K79 methylation and RNAPII eviction and degradation at replication fork–transcription complex collision sites [40].

## 2. Alterations of Chromatin Remodeler Complexes in Ovarian Cancer

### 2.1. ARID1A Alterations in Ovarian Cancer

About 25% of human cancers harbor mutations in at least one of 29 genes encoding SWI/SNF proteins [41]. Among them, *ARID1A* (BAF250a, B120, C1orf4, Osa1) is the most frequently altered. *ARID1A* mutations are particularly prevalent in gynecologic cancers (found in 10–60% of ovarian and endometrioid carcinoma cases) [42] and pre-malignant gynecological lesions, especially of endometrioid origin [43]. The highest *ARID1A* mutation rates are found in ovarian clear cell carcinoma (OCCC) and endometroid ovarian carcinoma (Table 2) suggesting its role in type I OC development and the potential for *ARID1A’s* use as a gynecologic cancer biomarker.

None of the previous research found any sufficient correlation between ARID1A protein expression loss and clinical features such as age, FIGO grade, and disease survival [60]. In a subset of microsatellite stable (MSS) endometrial cancer cases, ARID1A’s loss has been associated with a better prognosis and indicated as a potential prognostic biomarker [61]. Further investigation is highly needed to determine ARID1A’s potential as a gynecologic cancer biomarker.

Usually, *ARID1A* alterations are loss-of-function mutations such as large deletions, frameshift, or nonsense mutations that lead to the largest BAF complex subunit loss and inactivation of the SWI/SNF complex. However, *ARID1A* mutations are not sufficient to promote tumorigenesis alone. Tumors with *ARID1A* loss often harbor mutations of the PI3K/AKT pathway, such as inactivating *PTEN* or activating *PIK3CA* mutations. In mice models, separate *ARID1A* or PI3K/AKT pathway mutations promoted ovarian hyperplasia, while concomitant inactivation of *ARID1A* together with *PIK3CA* activating mutations induced tumors, similar to ovarian clear cell carcinoma (OCCC) [62].

ARID1A downregulation due to inactivating mutations has many downstream effects through epigenetic mark displacement, deregulating gene expression, and reduced protein interactions, primarily affecting DNA damage repair and signaling pathways (Table 3).

SWI/SNF complex regulates active H3K27ac histone mark at enhancer and promoter regions through interaction with p300/CBR histone acetyltransferase [63]. The acetylation of +1 nucleosome may be required for normal RNA polymerase II (RNAPII) pausing, a crucial step in effective transcription initiation [64]. Thus, ARID1A’s downregulation results in dysregulated expression of at least 99 target genes [82]. The impairment of ARID1A’s function due to inactivating mutations is not fully compensated by its homolog ARID1B. Most notably, one of the affected genes is tumor suppressor *TP53* [64]. *ARID1A* and *TP53* mutations are mutually exclusive in gynecologic malignancies, however, p53 is indirectly regulated by ARID1A. ARID1A is an HDAC6 deacetylase transcriptional repressor. Thus, the loss of ARID1A leads to HDAC6 reactivation, which causes p53 lysine 120 residue deacetylation and apoptosis inhibition [65].

ARID1A’s deficiency has also been linked with telomere impairment. In *ARID1A*-deficient cells, topoisomerase IIα (TOP2A) cannot interact with the SWI/SNF ATPase BRG1 (SMARCA4), which it needs to resolve catenanes that develop during replication and transcription [69]. Moreover, due to ARID1A insufficiency, TOP2A is not recruited to resolve R-loop sites and transcription-replication conflicts, which causes replication stress and increased genomic instability [70]. However, during mitosis, ARID1A deficiency results in cohesion protein STAG1 downregulation, and cancer cells with chromosome aberrations are eliminated during mitosis due to telomere cohesion deficiency [68]. ARID1A also downregulates promoters of telomerase reverse transcriptase; thus, *ARID1A* silencing increases telomerase activity, adding to the survival of cancer cells [71].

*ARID1A*’s mutational status in OC cells is also associated with reactive oxygen species (ROS) formation. *ARID1A* knockdown in OC cell lines increased intracellular ROS levels, as well as increased oxidative stress marker 8-hydrixyguanosine levels in OCCC patients with low ARID1A expression. Moreover, *ARID1A*-mutated cell lines were sensitive to ROS inductor elesclomol [76]. Another study found that ARID1A deficient cells have low antioxidant glutathione (GSH) levels and lower *SLC7A11* (cysteine transporter required by GSH) expression making them vulnerable to GSH and glutamate-cysteine ligase synthetase catalytic subunit (GCLC) inhibition and possible cancer treatment through increased ROS formation [75].

Telomere defects and DNA damage caused by *ARID1A* insufficiency leads to increased reliance on double-stranded DNA break (DSB) repair mechanisms [68]. Inactivation or loss of ARID1A or SWI/SNF complexes increases cell sensitivity to cisplatin and UV treatment through the impairment of multiple DSB repair mechanisms [74]. Most notably, ARID1A is involved in HR through DNA end processing (RPA and RAD51 loading), ATR activation, and G2-M cell-cycle arrest maintenance [72]. BAF factors, particularly ARID1A/B, are required for KU70/KU80 protein recruitment to DSB sites during NHEJ [73] as well as XPA accumulation at UV damage sites during NER (nucleotide excision repair) [74]. Moreover, ARID1A interacts with miss-match repair protein MSH2 and directs it to the damage site, starting the process of DNA damage repair [61]. The loss of functional ARID1A protein may induce DNA damage and decrease the cell’s ability to repair it.

In ovarian and endometrial cancers, *ARID1A* mutations often co-occur with PI3K/AKT pathway gene *KRAS*, *PIK3CA*, and *PTEN* alterations [77]. Additionally, ARID1A’s loss activates the pathway by *ANXA1* (AKT activator) upregulation and *PIK3IP1* (PI3K inhibitor) expression downregulation. In particular, *PIK3IP1* is regulated by ARID1A, suppressing EZH2 methyltransferase that inhibits *PIK3IP1* expression through the H3K27me3 epigenetic mark [83]. Therefore, targeting epigenetic regulators EZH2 (polycomb repressive complex (PRC2)), HDAC2 (PRC2 binding partner), and GSK126 (EZH2 inhibitor) are considered viable treatment options for *ARID1A* mutated tumors [66].

Besides its role in SWI/SNF, ARID members have also been reported to co-precipitate with members of E3 ubiquitin ligase. Specifically, ARID1B is linked with elongin C (EloC) through BC box motif and with the addition of cullin 2 and ROC1 form E3 ubiquitin ligase that targets H2B histone explicitly at lysine 120 for monoubiquitination. ARID1B is a paralog that shares most of its sequence with ARID1A. Thus, a similar E3 ubiquitin ligase complex may be formed with ARID1A too. Mutation and depletion of the ARID domain result in decreased ubiquitination of H2B, similar to VHL mutations, causing decreased ubiquitination of HIF1α in clear cell renal cell carcinoma (ccRCC) [80]. The two genes are often mutated together in ccRCC. The reduced H2B ubiquitination is responsible for reduced H3 histone lysine 79 di-methylation and decreased gene expression [84].

To sum up, ARID1A is highly involved in many pathways related to cancerogenesis (Figure 1), and its loss is associated with DNA damage repair, cell cycle, and apoptosis regulation through TP53 [65], possible ROS formation [75], telomere maintenance [68], and PI3K/AKT/mTOR pathway deregulation [66], providing ample opportunity for *ARID1A*-mutated OC treatment.

The frequent loss of function *ARID1A* mutations in cancers arising from endometriosis (endometrial tissue growth outside the uterus) makes SWI/SNF an attractive option for targeted therapy approaches. Although *ARID1A* has a mutually exclusive homolog *ARID1B*, and both *ARID1A* and *ARID1B* are often co-mutated in human cancers [85], the creation of specific inhibitors is challenging due to the 60% homology between ARID domains [24]. A more powerful therapeutic strategy is to target BRD2, which, together with ARID1A, is a member of BET (bromodomain and extra terminal domain) family proteins. BRD2 inhibitors reduce ARID1B and other SWI/SNF member expression in *ARID1A* mutant cancers [78].

Due to ARID1A’s involvement in DDR responses, *ARID1A* mutant cancers may be more susceptible to ATR blockade [86] and PARP inhibition [72]. Clinical trials involving ATR and PARP inhibition in OC patients are underway [87]. Although initially, ARID1A suppression in cancer cell line showed increased cisplatin sensitivity [74], ARID1A deficient ovarian cancer cells showed decreased chemosensitivity to cisplatin [88]. However, *ARID1A* mutational status may be a good predictor of immunotherapy outcomes: *ARID1A* deficient OC mouse model showed higher mutational burden, more tumor-infiltrating lymphocytes, and elevated levels of programmed cell death-ligand 1 (PD-1), providing an opportunity for immunotherapy [61]. A study found that patients with ARID1A deficiency that undergone anti-PD-1/PD-L1 immunotherapy had longer progression-free survival than patients with wild-type *ARID1A* tumors [89]. In another study involving *ARID1A* deficient OCCC cell lines, dasatinib, a tyrosine kinase inhibitor, was identified as a potential drug target. *ARID1A* mutated cell lines were sensitive to dasatinib treatment because of impaired cell cycle and YES1 (dasatinib target) deregulation [67].

### 2.2. Other SWI/SNF Alterations in Ovarian Cancer

Although *ARID1A* is the most frequently mutated SWI/SNF subunit coding gene, other SWI/SNF members, albeit less often, are also altered in OC. In particular, up to 90% of a rare form of OC, small cell ovarian carcinoma, hypercalcemic type (SCCOHT), a rhabdoid-like tumor has germline or somatic mutations of SWI/SNF ATPase domain BRG1 coding gene *SMARCA4* [90]. Just like ARID1A, the SMARCA4 is not only involved in gene expression through the primary SWI/SNF function as chromatin remodeler (*SMARCA4* mutations cause hyperactivation of PRC2 repressive complex) [43], but it also directly participates in DNA repair. BRG1 (SMARCA4) is recruited to DNA damage response (DDR) sites by PARP1 and is activated by deacetylation by SIRT1 to open the affected chromatin region in preparation for HR [91]. Moreover, BRG1 has been associated with γ-H2AX (a histone mark that indicates damaged DNA) formation, thus likely to promote DSB [92].

Meanwhile, other forms of OC, such as undifferentiated uterine or ovarian carcinoma and OCCC, are found to harbor alterations in other SMARC genes [93]. Most notably, other SMARC gene mutations are found in OCCC: *SMARCA4* in up to 5%, *SMARCA1* at 2%, *SMARCA2* at 1%, and *SMARCC1* at 2% [94]. Although *SMARCA2* mutations are rare, most SCCOHT cases present with dual inactivation of both BGR1 and BRM, though, unlike *SMARCA4*, *SMARCA2* is silenced epigenetically [95]. Similar to other rhabdoid tumors, SCCOHT may also harbor SWI/SNF core subunit *SMARCB1* (also known as INI1/SNF5/BAF47) mutations instead of *SMARCA4* [90]. Moreover, *SMARCB1* mutations are also found in other undifferentiated OC cases and myoepithelium-like tumors of the vulvar region [43]. Interestingly, SMARC coding SWI/SNF together with ARID1A is involved in oncogene *MYC* regulation: MYC is a transcriptional modulator of *SMARCB1*. However, ARID1A regulates MYC through binding with the *MYC* promoter region, and BRG1 (coded by *SMARCA4*) controls MYC partner MAX in a similar manner [81].

Curiously, in HGSOC, SMARC protein SMARCC1 was shown to be regulated on the protein level through methylation by coactivator-associated arginine methyltransferase 1 (CARM1/PRTM4), which is overexpressed in HGSOC. The methylation causes BAF complex eviction from certain loci, including promoters of c-Myc pathway genes, thus altering gene expression [96].

Although the loss of function mutations in SWI/SNF complex encoding genes occur in more than 20% of human cancers, some amplifications of genes coding for SWI/SNF subunits are also found. Most notably, a scaffolding protein, coded by *ACTL6A* and ncBAF subunit gene *BRD9*, are overexpressed in most cancers, including OC. The amplification of these genes co-occurs with notable oncogene *TP63* amplification, suggesting an oncogenic mechanism of OC [97]. However, *ACTL6A* amplification has also been associated with follicle-stimulating hormone regulation through PIK3/AKT pathway. *ACTL6A* increase leads to enhanced glycolysis due to increased expression of phosphoglycerate kinase 1 (PGK1) [98]. Moreover, ACTL6A is involved in regulating cisplatin resistance through DNA repair enhancement [99].

Overall, SWI/SNF gene alterations, especially inactivating mutations, are frequent events in OC development. Besides *ARID1A*, SMARC genes coding ATPase and regulatory subunits are frequently altered as well. An investigation is needed on whether these genes could serve as novel biomarkers for OC.

### 2.3. ISWI Alterations in Ovarian Cancer

Similarly, to SWI/SNF, ISWI ATPase domains are also encoded by SMARC genes that frequent alterations in OC. Most notably, ISWI ATPase SNF2L (*SMARCA1*) mutations have been linked with OCCC [94]. Although SNF2H is the dominant ISWI ATPase, SNF2L has been described as a regulator of specific gene expression during differentiation. Crucially, SNF2L has been associated with ovarian development and meiotic progression of germ cells during its differentiation [100]. SNF2L interacts with progesterone receptor and steroidogenic acute regulatory protein in ovarian granulosa cells to promote differentiation of the ovary [101]. *SMARCA1* deletion has been associated with increased apoptosis through caspase activator Apaf-1 expression upregulation [102] and enhanced proliferation and migration due to WNT signaling regulation in various cancer cells [103].

Depending on which ATPase it binds, ISWI protein remodeling and spacing factor 1 (coded by *RSF1*, also known as BAZ1A, HBXAP), forms either RSF-1 or RSF-5 CRC [27]. Immunohistochemical analysis showed that ISWI ATPase SNF2H (*SMARCA5*) is overexpressed in OC tissues together with its binding partner RSF1 which is also often upregulated in various tumors, including OC [104]. RSF1 is an essential interphase centromere protein that maintains chromosome stability and protein homeostasis but also has roles in DSB and transcription regulator functions through its interactions with HDAC1, CENP-A, ATM, SNF2H, cyclin E1, CBP, NF-κB, BubR1, and many other cancer-related proteins [105]. The RSF-1 complex protein expression correlates with cancer stage and poor clinical outcomes in OC patients [106]. The upregulation of RSF-1 and SMARCA5 expression leads to increased double-stranded breaks (DSB) and subsequent DDR [107]. However, in HGSOC cells that are almost invariably deficient in *TP53*, the DNA damage caused by RSF-5 upregulation goes unsolved [108]. Moreover, RSF1 has been co-immunoprecipitated with cyclin E1, which is also overexpressed in OC cells. In HGSOC cells, cyclin activation by RSF1 leads to proliferation and tumor formation via activation of cyclin E1-associated kinase (CDK2) [109]. RSF1 overexpression has also been linked with chemoresistance in OC. RSF1 overexpression in SKOV3 OC cell line activated NF-κB transcription factor and subsequent gene expression changes resulting in decreased apoptosis (*CFLAR*, *XIAP*, *BCL2*, *BCL2L1*) and inflammation (*PTGS2*) gene expression adding to paclitaxel resistance in ovarian cells [110]. Thus, RSF1 overexpression may serve as a prognostic biomarker for OC patients.

In addition to *RSF1*, amplification of *CHARC1* and *RBBP4* has also been identified in HGSOC, along with *RSF1* and *RBBP4* gene fusions. CHRAC complexes (*CHRAC1*, *RSF1*, *POLE3*) are known oncogenic drivers regulating proliferation and survival in many cancer types [27]. In cisplatin-resistant OC cell lines, *CHRAC1* expression was found to be increased due to *miR-512-3p* downregulation and was related to enhanced proliferation, invasion, migration, and apoptosis [111]. RBBP4 is not the only ISWI family NURF CRC component, but also a member of other transcription regulation complexes, such as MuvB, which is a part of DREAM, MMB, and FoxM1-MuvB regulatory complexes. During different cell cycle phases, MuvB acts either as a transcriptional repressor or activator. DREAM transcriptional complex has been associated with carboplatin chemoresistance, and depletion of DREAM proteins might prove a viable option for OC treatment [112].

### 2.4. CHD Family Alterations in Ovarian Cancer

CHD5 is the main CHD family protein linked with cancer [34]. Although mutations in the *CHD5* gene are detected in OC, *CHD5* is also downregulated by increased promoter methylation in OC patient samples [113]. In one study analyzing CHD gene promoter methylation in various cancer cell lines, out of all CHD genes, *CHD5* promoter was methylated the most often, although the OC cell line (MDAH2774) showed no increase in methylation [114]. *CHD5* polymorphism (rs9434741) has also been associated with endometriosis, an OC precursor [115], however, the clinical significance of this variant is not known. In alignment with *CHD5* increased mutation and methylation, *CHD5* mRNA expression was found to be downregulated by at least 2-fold in 41% (32/72) of invasive epithelial ovarian carcinomas in comparison with 12 controls and correlated with shorter disease-free survival times [116].

Besides *CHD5*, CHD CRCs are also found to be dysregulated in ovarian and other cancers and have a role in chemoresistance. *CHD8* overexpression has been linked with poor survival in both HGOSC and endometrial cancer cases [117]. CHD8 binds to β-catenin and β-catenin regulated gene promoters to act as a negative regulator of WNT, a prominent pathway OC [118]. Similarly, CHD4 overexpression also correlated with poor survival and was significantly higher in platinum-resistant HGSOC cases. CHD4 was found to induce chemoresistance through multi-drug transporter MDR1 expression regulation [119]. Other NuRD factors, besides CHD, are also found to confer platinum resistance: methyl-CpG binding domain protein 3 (MBD3) is required for platinum resistance through H3K27ac mark regulation at enhancer regions and altered expression of JAK/STAT and FGF signaling pathway genes [120]. Meanwhile, another NuRD protein, MTA1 overexpression in OC tissues correlates with cancer stage and grade, while in the OVCAR-3 cell culture model MTA1 upregulation reduced estrogen receptor β expression and cytokine GRO (CXCL1) upregulation possibly explaining MTA1 role in OC progression and metastasis [121].

### 2.5. INO80 in Ovarian Cancer

Although several INO80 subunits are overexpressed in melanoma [122], cervical [123], and non-small cell lung cancer [124], according to TCGA data analysis [125,126] the only INO80 gene significantly amplified in OC patients is *ACT6LA*, shared with SWI/SNF family remodelers. *ACT6LA* amplification affects multiple CRCs and causes platinum resistance in OC [99]. However, other INO80 family domains are found to be significantly altered in OC cell models. TRRAP (transformation/transcription domain-associated protein)—a TIP60 remodeler/histone acetyltransferase complex adaptor protein is overexpressed in OC A2780 sphere cultures and was found to govern stemness marker NANOG expression and OC cell proliferation [127]. Yin yang 1 (YY1)—a non-conserved human INO80 family subunit [38] in OC cell line models, has been associated with chemoresistance through upregulation of lncRNA *PART1*, which targets *miR-512-3p* and causes *CHRAC1* (another CRC gene) upregulation and cisplatin-resistant OC cell proliferation and migration [111]. In OC patients and cell lines YY1 itself was found to be suppressed by *miR-381*, which is downregulated in OC tissues [128]. In another study, higher YY1 protein expression in OC was associated with reduced survival [129].

In 495 Chinese epithelial OC patient study analyzing INO80 module gene *RUVBL1*, single nucleotide polymorphisms rs1057156 (A>G) and rs149652370 (A>G) were significantly associated with poor overall survival. The rs1057156 (A>G) variant was found to lower *miR-4294* binding affinity causing increased *RUVBL1* expression in OC cells [130]. In OC patients, RUVBL1/2 has also been found to be crucial for mTOR inhibitor sensitivity as it mitigates cell stress caused by mTORC1-Myc-DNA damage axis [131].

## 3. Conclusions and Perspectives

Overall, at least 18 different CRC subunits are affected in OC. While SWI/SNF genes are mostly affected by inactivating mutations, amplification and epigenetic changes also affect CRC coding genes (Table 4). These alterations, especially mutations and changes in gene expression, could be helpful as prognostic, predictive, and diagnostic markers for gynecologic malignancies (Figure 2). Although OC is a highly heterogeneous disease, a better understanding of the mutational landscape of individual tumors will help to choose the right therapy for each patient. Unlike mutations in classical cancer-related genes such as *TP53*, *BRCA1/2*, *MYC* or *RAS*, CRCs and their involvement in cancer pathways are known only for a few decades [132], thus, are much less studied. Arguably, SWI/SNF is the main CRC of interest, it was the first CRC discovered, it’s involved in a wide variety of cellular functions and shows high mutation load in *ARID1A* and other genes in OC and other cancers, providing ample ground for possible therapeutic options [66]. However, other CRCs are also deregulated in OC, and may also be useful for OC diagnostics and treatment.

Recently identified numerous mutations and amplifications in chromatin remodeling genes open a possibility for new therapeutic approaches. Of note, FDA approval of EZH2 inhibitor tazemetostat therapy for follicular lymphoma shows a promising step in epigenetic therapy application, and increases hope that the same inhibitor could be used for SWI/SNF deficient HGSOC, OCCC, endometrial OC, or SCCOHT patient treatment [135]. Phase II studies for this and other SWI/SNF deficient cancer therapies are already underway [41]. Most of the current research on CRCs only considers separate CRC protein or gene dysregulation effects on OC cells or small groups of patients, however, a broader picture of how CRCs are affected by these particular mutations or amplifications and how this dysregulation affects OC patients is greatly needed. A better understanding of CRCs will aid in the development of new cancer treatment options guided by biomarker-based OC diagnostics.

## Figures and Tables

**Figure 1 ijms-23-13670-f001:**
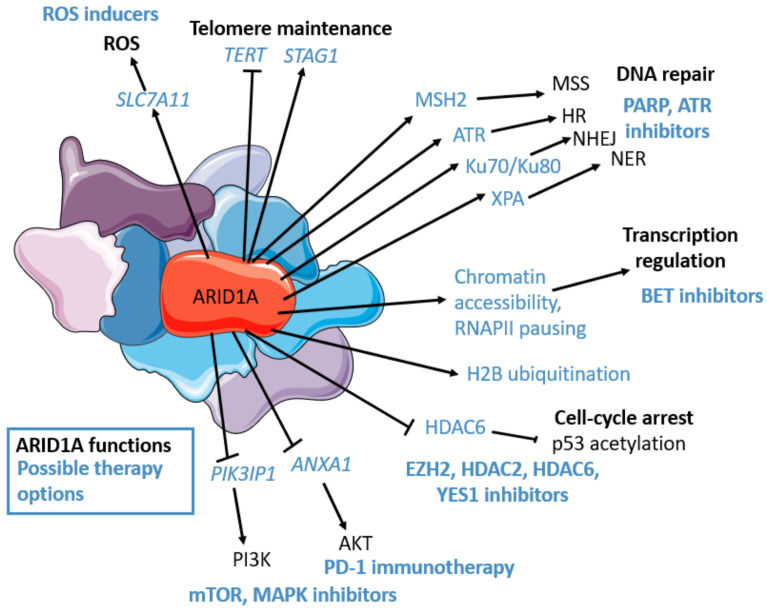
ARID1A’s functions (in black): transcriptional regulation (chromatin accessibility, epigenetic mark placement, RNAPII pausing regulation), PI3K/AKT/mTOR pathway regulation, cell-cycle regulation (p53 deacetylation through HDAC6), DNA repair: Homologous repair (HR) (RPA and RAD51 loading, ATR interaction, and activation), non-homologous end joining (NHEJ) (Ku70/Ku80 interaction), nucleotide excision repair (NER) (XPA interaction), miss match repair (MSS) (MSH2 interaction), Telomere regulation (STAG1 maintenance, *TERT* downregulation), oxidative stress repression (though *SLC7A11* maintenance). ARID1A’s possible therapeutic options (in blue): cisplatin, PD-L1, PARP, ATR, YES1, BET (BRD2), HDAC6, HDAC2, EZH2, PI3K, AKT inhibitors, mTOR, MAPK inhibitors, ROS inducers. The figure was generated using Servier Medical Art, provided by Servier, licensed under a Creative Commons Attribution 3.0 unported license.

**Figure 2 ijms-23-13670-f002:**
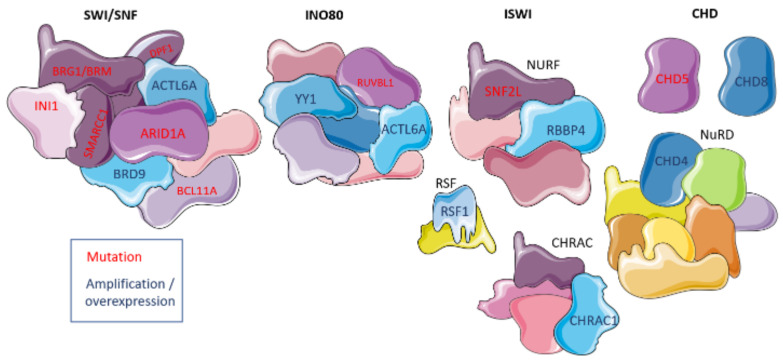
Chromatin remodeling complexes and their involvement in ovarian cancer. Deregulated domains are shown in either purple with red text denoting mutations in coding genes or blue with dark blue text denoting expression up-regulation. The figure was generated using Servier Medical Art, provided by Servier, licensed under a Creative Commons Attribution 3.0 unported license.

**Table 1 ijms-23-13670-t001:** SWI/SNF main subunits and coding genes.

SWI/SNF Subunits	Coding Gene	Protein Names
	*SMARCA4*	BRG1
Helicases/ATPases	*SMARCA2*	BRM
	*ACTB*	β-actin
ATPase cap subunit	*ACTL6A/B*	BRAF53A/B
	*BCL7A/B/C*	
BAF specific	*SS18/SS18L1*	
	*BCL11A/B*	
	*SMARCB1*	BAF47/INI1
	*SMARCE1*	BAF57
Common core subunit	*SMARCC1*	BAF155
	*SMARCD1/2/3*	BAF60A/B/C
	*SMARCC2*	BAF170
	*ARID1A/B*	BAF250A/B, OSHA1/2
cBAF specific	*DPF1/2/3*	BAF45B/D/C
	*ARID2*	BAF200
	*PBRM1*	BAF180
PBAF specific	*PHF10*	BAF45A
	*BRD7*	
	*GLTSCR1/L*	BIRCA/L
ncBAF specific	*BRD9*	

**Table 2 ijms-23-13670-t002:** *ARID1A* mutation frequency in gynecologic diseases.

Gynecologic Disease	*ARID1A* Mutation Frequency	References
Endometriosis	10% (9/107)	[44]
	2% (2/101)	[45]
Low-grade endometrioid endometrial cancer	46.7% (129/276)	[46]
High-grade endometrioid endometrial cancer	60% (18/30)	[46]
Serous endometrial cancer	10.8% (4/37)	[46]
	23.8 % (10/42)	[46]
Endometrial carcinosarcoma	12% (7/57)	[47]
	30% (10/33)	[48]
Endometrioid endometrial cancer	40% (10/25)	[49]
	68% (11/16)	[50]
	36% (40/122)	[51]
	45% (9/20)	[52]
Endometrioid ovarian cancer	43% (6/14)	[53]
	36% (8/22)	[54]
Granulosa cell ovarian cancer	0% (0/5)	[53]
	0% (0/76)	[48]
	0% (0/36)	[52]
High-grade serous ovarian cancer	1% (1/98)	[55]
	48% (11/23)	[54]
Low-grade serous ovarian cancer	8.5% (6/71)	[56]
	33% (2/6)	[52]
Mucinous ovarian cancer	50% (2/4)	[53]
	46% (55/119)	[48]
	42% (17/41)	[52]
	69.7% (69/99)	[57]
Clear cell ovarian cancer	57% (24/42)	[58]
	20% (1/5)	[53]
	14% (5/37)	[54]
Seromucinous borderline ovarian cancer	14% (4/27)	[59]

**Table 3 ijms-23-13670-t003:** Effects of *ARID1A* loss of function mutations and opportunities for cancer therapy.

Affected Cell Processes	Result	Actionable Therapy Approach	References
**Gene expression changes:**			
p300/CRB histone acetyltransferase interaction	Due to lost p300/CRB interactions with ARID1A, changes in H3K27ac marks lead to many gene expression changes		[63]
RNAPII pausing	Loss of normal RNAPII pausing may be due to reduced promoter acetylation and results in decreased gene expression		[64]
HDAC6 deacetylase activation, p53 deacetylation	HDAC6 is activated by *ARID1A* reduction. In turn, p53 K120 is deacetylated (which inhibits apoptosis)	HDAC6 inhibitor	[65]
ARID1B	As an ARID1A homolog, ARID1B is left to restore most of the functions	BRD2 (BET, required for *ARID1B* transcription) inhibitors	[66]
YES1 deregulation	In *ARID1A* deficient cell line model treated with dasatinib, YES1 was identified as the main target gene	YES1 inhibitor (dasatinib)	[67]
**Telomere/cell division:**			
STAG1 (cohesin protein) reduction	Reduced STAG1 leads to decreased mitotic telomere cohesion		[68]
TOP2A (topoisomerase) interaction loss	TOP2A interactions with SWI/SNF complex ATPase BRG1 are lost/reduced. This causes replication stress through reduced sister chromatid decatenation and anaphase bridge formation during mitosis		[69,70]
*TERT* (telomerase reverse transriptase) upregulation	*TERT* promoter is upregulated in *ARID1A* deficient cells (increased survival)	ATR inhibitor	[71]
**DNA damage reparation:**			
HR (homologous DSB repair)	ARID1A interacts and activates ATR and is required for DNA end processing (RPA and RAD51 loading), as well as G2-M cell-cycle arrest maintenance	PARP inhibitor	[72]
NHEJ (non-homologous end joining)	ARID1A/B are required for KU70/KU80 protein recruitment to DSB	cisplatin	[73]
MMR (miss match repair)	ARID1A may interact and direct MSH2 to MMR sites	PD-L1 inhibitor	[61]
NER (nucleotide excision repair)	ARID1A/B required for XPA (NER protein) requirement		[74]
ROS (reactive oxygen species) formation	Reduces *SLC7A11* levels to impair antioxidant GSH production	ROS inducers (e.g., HSP90 inhibitor, elesclomol)	[75,76]
**PI3K/AKT/mTOR pathway:**			
PI3K/AKT/mTOR regulation	Mutually inclusive *KRAS*, *PIK3CA*, and *PTEN* mutations		[77]
ANXA1 (AKT activator) upregulation	Upregulation by *ARID1A* loss	AKT inhibitor	[78]
PIK3IP1(PI3K inhibitor) downregulation	Downregulation by *ARID1A* loss (through EZH2 methyltransferase activation)	PI3K, EZH2, HDAC2 inhibitors	[79]
**Protein interactions:**			
E3 ubiquitin ligase interaction	ARID1 forms E3 ubiquitin ligase, and mutations result in decreased ubiquitination of H2B histones		[80]
MYC interaction	Interaction with and regulation of MYC		[81]

**Table 4 ijms-23-13670-t004:** Overview of CRC alterations other than *ARID1A* in ovarian cancer. OC—ovarian cancer, SCCOHT—small cell ovarian carcinoma, hypercalcemic type, OCCC—Ovarian clear cell carcinoma, HGSOC—High-grade serous ovarian cancer.

Chromatin Remodeler Complex	Gene	Change	Type of OC	References
			SCCOHT	[133]
SWI/SNF	*SMARCA4*	Inactivating mutation	OCCC	[94]
			Dedefenrentiated OC	[93]
SWI/SNF	*SMARCB1*	Inactivating mutation	SCCOHT	[95]
		Epigenetic silencing, downregulation	SCCOHT	[134]
SWI/SNF	SMARCA2	Inactivating mutation	OCCC	[94]
SWI/SNF, ISWI	*SMARCA1*	Inactivating mutation	OCCC	[94]
SWI/SNF	*BCL11A*	Inactivating mutation	OCCC	[94]
SWI/SNF	*DPF1*	Inactivating mutation	OCCC	[94]
		Inactivating mutation	OCCC	[94]
SWI/SNF	*SMARCC1*	Protein methylation by upregulated CARM1	HGSOC	[96]
SWI/SNF, INO80	*ACTL6A*	Amplification	Non-specified OC	[97]
SWI/SNF	*BRD9*	Amplification	Non-specified OC	[97]
ISWI	*RSF1*	Amplification, gene fusion	HGSOC	[27,104]
ISWI	*CHARC1*	Amplification	HSGOC	[27]
ISWI	*RBBP4*	Amplification, gene fusion	HSGOC	[27]
CHD	*CHD5*	Inactivating mutation, promoter methylation	Non-specified OC	[113,116]
CHD	*CHD8*	Amplification	HSGOC	[117]
CHD	*CHD4*	mRNA overexpression	Non-specified OC	[119]
INO80	*RUVBL1*	Polymorphisms	Epithelial OC	[130]
INO80	*YY1*	Protein overexpression	Epithelial OC	[129]

## Data Availability

Not applicable.

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
