# Peer review of "The Emerging Role of Chromatin Remodeling Complexes in Ovarian Cancer"

_ijms, 2022, doi:10.3390/ijms232213670_

Round 1

Reviewer 1 Report

The review by Vaicekauskaite and colleagues tackles the topic of chromatin remodelers in ovarian cancer. It is a well written review, with extensive details and maybe somewhat too long. It would be interesting if the authors could propose at the end a model (i.e. figure), of how chromatin remodeling complexes affect and/or control ovarian cancer, which would summarize all the previous information.   

Reviewer 2 Report

This review by Vaicekauskaité et al. covers the topic of genetic defects in chromatin remodeling genes as a cause of ovarian cancer. It provides comprehensive and dense information to interested readers and draws attention to potential targets for treating these tumors as well as deeping into uncharacterized disease mechanisms that could be applied in personalized medicine.

Author Response

Thank you for your kind review.

Reviewer 3 Report

This review provides an overview of the alterations of members of chromatin remodeling complexes observed in ovarian cancer patients/cell lines with the aim to summarize the state of the art and suggest new target to be investigated for the development of innovative treatment.

The manuscript is well organized, the authors firstly described the ovarian cancer subtypes, then all the chromatin remodeling complexes and their functions (SWI/SNF complex, ISWI complex, CHD molecules and INO80 family) and finally the chromatin remodeling complexes deregulation in ovarian cancer with particular interest on ARID1A.

Only some MINOR points need to be addressed:

1) Line 205: remove “cancer” after OC

2) Line 335: change “similar” with “similar”

3) Line 366: change “ether” with “either”

4) Line 382: change “NF-B” with “NF-kB”

5) Line 411: change “promotors” with “promoters”

6) Change “O verview” in the figure legend of table 4 with “overview”

7) The sentences in lines 356/357 – 389 do not seem correct, please check

Author Response

Thank You for Your kind review. We have made the suggested changes in the manuscript:

1) Line 205: removed “cancer” after OC

2) Line 335: changed “simmilar” with “similar”

3) Line 366: changed “ether” with “either”

4) Line 382: changed “NF-B” with “NF-kB”

5) Line 411: changed “promotors” with “promoters”

6) Changed “O verview” in the figure legend of table 4 with “Overview”

7) The sentences in lines 356/357 – 389 changed from:

Arguably, due to its high mutability in many cancers, the most well-studied CRC is SWI/SNF, and the studies have shown its involvement in a wide variety of cellular functions such as transcription regulation, DNA repair, cell cycle regulation, and others, providing ample ground for possible therapeutic options \cite{Caumanns2018}. However, other CRCs are also required for many of these cancer-related processes, although the details of the underlying mechanisms are yet to be discovered \cite{Clapier2021}.

To

Arguably, SWI/SNF is the main CRC of interest, it was the first CRC discovered, it’s involved in a wide variety of cellular functions and shows high mutation load in ARID1A and other genes in OC and other cancers, providing ample ground for possible therapeutic options \cite{Caumanns2018}. However, other CRCs are also deregulated in OC, and may also be useful for OC diagnostics and treatment.

Thank You for Your revisions.